# Two Students: Enabling Uncertainty Quantification in Federated Learning Clients

**Cristovão Freitas Iglesias Jr**$^*$
University of Ottawa

**Sidney Alves de Outeiro**
Federal University of Rio de Janeiro

**Claudio Miceli de Farias**
Federal University of Rio de Janeiro

**Miodrag Bolic**
University of Ottawa

## Abstract

Federated Learning (FL) is a paradigm where multiple clients collaboratively train models while keeping their data decentralized. Despite advancements in FL, uncertainty quantification (UQ) on the client side remains poorly explored. Existing methods incorporating Bayesian approaches in FL are often resource-intensive and do not directly address client-side UQ. In this paper, we propose the 2S (two students) approach to address this gap. Our approach distills a Bayesian model ensemble (BME) into two student models: one focused on accurate predictions and the other on uncertainty quantification. The 2S approach also includes a novel truncation filter that uses credible intervals to selectively aggregate client models, mitigating the impact of non-i.i.d. data. Through empirical validation on a regression task, we demonstrate that the 2S approach enables effective and scalable UQ on the client side, providing robust and reliable updates across decentralized data sources.

## 1 Introduction

Federated Learning (FL) is a machine learning approach in which distributed clients with isolated data work together to solve a learning problem, typically under the guidance of a central server [15, 17]. Due to the need to ensure reliable and robust decision-making by accurately assessing the variance in the client model predictions across decentralized and heterogeneous data sources, FL requires uncertainty quantification (UQ) [42, 22]. However, the literature about UQ in FL is very limited [36]. Traditional FL methods like federated averaging (FedAvg) [15] and its variants primarily [40] focus on model aggregation across distributed clients but often neglect the explicit incorporation of uncertainty in model predictions [22]. Some recent works have started to integrate Bayesian learning principles into FL framework to capture uncertainty, but they are typically resource-intensive and do not directly address the need for UQ on the client side [5, 6]. For example, Bayesian model ensemble (BME) is used as a method for aggregating the locally trained models from different clients on the server [6, 36]. The BME in FL offers robustness and better uncertainty quantification but makes more complex the process of obtaining a single, definitive global model (easy to use in practice) that reproduces the predictive mean and uncertainty rules of BME [5]. **Motivation and problem specification**. As in FL, where models are trained on data distributed across many clients, estimating uncertainty becomes challenging, particularly due to the limited computational resources available on client devices. Therefore, "cost-effective" uncertainty quantification in FL is still an open problem [15, 22, 42, 5, 36], where "cost-effective" means allowing clients to perform uncertainty quantification without requiring extensive computational resources or complex procedures.

---

$^*$cfrei096@uottawa.ca | https://cristovaoiglesias.github.io/personalwebsite/

Workshop on Bayesian Decision-making and Uncertainty, 38th Conference on Neural Information Processing Systems (NeurIPS 2024).

Consequently, the following question remains unanswered: *How can we obtain global parameters directly from BME to enable a cost-effective uncertainty quantification on the client side?* **Our contribution**. To address the above problem, we propose the two students (2S) approach. It involves distilling (knowledge distillation [34]) a BME into two student models (suitable for deployment on various devices) within a FL framework. The first student model, $\boldsymbol{w}_{pred}$, is designed to generate accurate predictions by mimicking the predictive mean of the ensemble. The second student model, $\boldsymbol{w}_{uq}$, focuses on uncertainty quantification by learning from the credible intervals (CIs) generated by the ensemble, which represent the uncertainty in predictions. Our empirical experiments using a classical regression problem show that the 2S approach can enable an efficient and cost-effective uncertainty quantification on the client side while filtering out unreliable client models (through a truncation filter) to ensure robust and consistent updates in a decentralized, non-i.i.d. data environment. The code and data used in this work are available in the GitHub repository [2] to facilitate reproducibility.
**Advantages.** (i) To the best of our knowledge, there is no prior work that combines the idea of using credible intervals from Bayesian ensemble predictions to filter out drifted client models in FL (see the related work in Appendix C). While there are works that focus on distillation or ensemble learning in federated settings, the explicit use of prediction uncertainty to selectively aggregate models represents a unique contribution. (ii) Moreover, by distilling the ensemble uncertainty into $\boldsymbol{w}_{uq}$, the 2S enables a cost-effective UQ by offloading the complexity to the server, allowing clients to operate with a lightweight model that still offers reliable uncertainty estimates. This reduces the complexity of performing UQ directly on client devices. This approach is efficient and scalable, making it suitable for deployment on resource-constrained client devices.

## 2 Background

FedAvg is a standard approach in FL. During the **client training stage**, each client $i$ has its own local dataset $D_i = \{(\boldsymbol{x}_n, \boldsymbol{y}_n)\}_{n=1}^{N_i}$, where $\boldsymbol{x}_n \in \mathbb{R}^v$ and $\boldsymbol{y}_n \in \mathbb{R}^d$. The current global model weights are denoted by $\bar{\boldsymbol{w}} \in \mathbb{R}^h$. Each client initializes their local model weights $\boldsymbol{w}_i \in \mathbb{R}^h$ with $\bar{\boldsymbol{w}}$ and performs stochastic gradient descent (SGD) for $K$ steps with step size $\eta_l$. The update rule for the local model $\boldsymbol{w}_i$ is $\boldsymbol{w}_i \leftarrow \boldsymbol{w}_i - \eta_l \nabla \ell(B_k, \boldsymbol{w}_i)$ where $\ell$ is the loss function and $B_k$ is the mini-batch sampled from $D_i$ at the $k$-th step. After the set of clients $S$ have updated their local models $\boldsymbol{w}_i$, we have the **model aggregation stage**. The server aggregates these models $\{\boldsymbol{w}_i; i \in S\}$ into a single global model by performing a weighted average: $\bar{\boldsymbol{w}} \leftarrow \sum_i \frac{|D_i|}{|D|} \boldsymbol{w}_i$ where $|D| = \sum_i |D_i|$ is the total size of all clients' data. This updated global model $\bar{\boldsymbol{w}}$ is then sent back to the clients for the next round of training. The process repeats for $R$ rounds. FEDAVG works well when the data $D_i$ on each client is i.i.d. relative to the aggregated data $D$ [6, 15]. However, in practice, the data is often non-i.i.d., which can degrade the performance of FEDAVG, leading to a global model $\bar{\boldsymbol{w}}$ that may drift away from the ideal model $\boldsymbol{w}^* \in \mathbb{R}^h$ (the model trained centrally with all the data). One approach to mitigate model drift is to perform **Bayesian inference** [2, 32, 6, 11] by integrating the outputs of all possible models $\boldsymbol{w}$ according to the *predictive posterior*: $p(\boldsymbol{y}|\boldsymbol{x}; D) = \int p(\boldsymbol{y}|\boldsymbol{x}; \boldsymbol{w}) p(\boldsymbol{w}|D) \, d\boldsymbol{w} \approx \frac{1}{M} \sum_{m=1}^M p(\boldsymbol{y}|\boldsymbol{x}; \boldsymbol{w}^{(m)}), \boldsymbol{w}^{(m)} \sim p(\boldsymbol{w}|D)$, rather than relying on a single point estimate. As that integral is intractable, it is approximated using M Monte Carlo samples $\boldsymbol{w}^{(m)}$ (global models) from the *posterior* $p(\boldsymbol{w}|D)$. This is the BME approach which can mitigate the model drift by leveraging the diversity in the sampled global models [6, 5].

## 3 2S approach

**Model aggregation in server.** Obtaining samples from the true posterior $p(\boldsymbol{w}|D)$ is virtually impossible, requiring the use of an *implicit sampling distribution* $q(\boldsymbol{w})$ to approximate $p(\boldsymbol{w}|D)$ [18, 11, 26]. The ensemble prediction in this context refers to the aggregated output from multiple global models sampled from the *approximate posterior* $q(\boldsymbol{w}) \approx p(\boldsymbol{w}|D)$. Therefore, given a new input $\boldsymbol{x}^*$, we obtain the *approximate predictive posterior* as, $\hat{p}(\boldsymbol{y}^*|\boldsymbol{x}^*, D) \triangleq \frac{1}{M} \sum_{m=1}^M p(\boldsymbol{y}^*|\boldsymbol{x}^*, \boldsymbol{w}^{(m)})$, where $\boldsymbol{w}^{(m)} \sim q(\boldsymbol{w})$, [11, 25]. The quality of the $\hat{p}(\boldsymbol{y}^*|\boldsymbol{x}^*, D)$ depends on the number of samples M and the method employed to generate $q(\boldsymbol{w})$ [11, 25, 26]. Furthermore, the uncertainty of the $\hat{p}(\boldsymbol{y}^*|\boldsymbol{x}^*, D)$, quantified by the credible intervals (CI), reflects the variance (spread level) in the aggregated output from multiple global models $\boldsymbol{w}^{(m)}$ (ensemble output). There are several ensemble

[2]https://github.com/cristovaoiglesias/2S

methods for generating $q(\boldsymbol{w})$, such as bagging, and negative correlation learning [9, 25, 26]. We propose using the client training process as a method to generate the implicit sampling distribution of global models, $q(\boldsymbol{w}) = \text{ClientTraining}(\boldsymbol{w}_{\text{pred}}, D_i)$. Here, $\boldsymbol{w}_{\text{pred}}$ represents a global model that reproduces the ensemble prediction rule. In this context, the client training process defines $q(\boldsymbol{w})$, and the trained models $\{\boldsymbol{w}_i; i \in S\}$ obtained from the training process serve as samples from $q(\boldsymbol{w})$. In FL, each client typically has its own local dataset $D_i$. As a result, the locally trained models $\boldsymbol{w}_i$ are likely to capture different aspects of the global data distribution [18]. This inherent diversity among trained models from clients can serve as a natural sampling mechanism to approximate $p(\boldsymbol{w}|D)$ [11]. Using these trained models allows the ensemble to capture a wide range of possible model parameters, reflecting the variability and uncertainty inherent in the data. Furthermore, studies such as those by Lakshminarayanan et al. [18] and Malinin et al. [26] demonstrate the effectiveness of using ensembles to approximate $p(\boldsymbol{w}|D)$ in centralized settings. By extending these ideas to the decentralized setting of FL, where diverse client models are naturally available, we can leverage the same underlying principles of uncertainty quantification and robustness. See the extended justification in Appendix B.1. As described before, in FL, a drifted model can be generated when the client training process is performed with non-i.i.d. data. The inclusion of drifted models $\boldsymbol{w}_i^{drift}$ in the ensemble of models $\{\boldsymbol{w}^{(m)}\}_{m=1}^M$ does not affect the mean of ensemble predictions $\frac{1}{M}\sum_{m=1}^M p(\boldsymbol{y}^*|\boldsymbol{x}^*, \boldsymbol{w}^{(m)})$ [28, 10]. However, it increases the uncertainty, which increases as more drifted models are included in the ensemble [26]. To solve this issue, we propose utilizing credible intervals to dynamically truncate distribution $q(\boldsymbol{w})$ and filter client models. We named this approach the truncation filter, and it is similar to sampling from a truncated distribution. To identify drifted models, we compute the credible intervals $\{[\boldsymbol{L}_{CI}^{(q)}, \boldsymbol{U}_{CI}^{(q)}]\}_{q=1}^Q$ for the ensemble of predictions $\{\{\hat{\boldsymbol{y}}_q^{(m)}\}_{q=1}^Q\}_{m=1}^M$ obtained with in-domain inputs $\bar{\boldsymbol{x}} = \{\bar{\boldsymbol{x}}_q\}_{q=1}^Q$, where each $\bar{\boldsymbol{x}}_q \in \mathbb{R}^v$ is part of an i.i.d. dataset $D_{init} = \{\bar{\boldsymbol{x}}_q, \bar{\boldsymbol{y}}_q\}_{q=1}^Q$ with $\bar{\boldsymbol{y}}_q \in \mathbb{R}^d$. Then, a trained model $\boldsymbol{w}_i$ is not considered drifted $\boldsymbol{w}_i^{drift}$ if a significant proportion $(\delta)$ of its predictions $\boldsymbol{y}' = \{\boldsymbol{y}'_q\}_{q=1}^Q$ (given $\bar{\boldsymbol{x}}$) falls within $\{[\boldsymbol{L}_{CI}^{(q)}, \boldsymbol{U}_{CI}^{(q)}]\}_{q=1}^Q$. If $\delta$ is below a certain threshold $\gamma$ (the truncation filter threshold), the model is not included in the ensemble. Specifically, $\boldsymbol{L}_{CI}^{(q)} \in \mathbb{R}^d$ and $\boldsymbol{U}_{CI}^{(q)} \in \mathbb{R}^d$ represent the lower and upper bounds of the credible interval computed for the ensemble of predictions $\{\hat{\boldsymbol{y}}_q^{(m)}\}_{m=1}^M$ at the input $\bar{\boldsymbol{x}}_q$. Therefore, $\delta < \gamma$ suggests that a client model was trained on data that is not representative of the overall i.i.d. data distribution seen by the ensemble. Including such a model in the ensemble would increase the uncertainty, thus signaling the presence of model drift. By selecting only those models whose a significant proportion of predictions fall within the credible intervals, we ensure that the global model is constructed from client models whose data distributions are aligned with the overall ensemble thereby minimizing the impact of model drift due to non-i.i.d. data.

**Cost-effective UQ in clients.** We must translate the prediction rule of the ensemble of models $\{\boldsymbol{w}^{(m)}\}_{m=1}^M$ into the global model $\boldsymbol{w}_{pred}$ to send back to the clients to continue client training. To this end, we use knowledge distillation [6, 34, 20] to transfer knowledge from a teacher model (the ensemble) to a student model ($\boldsymbol{w}_{pred}$). We use unlabeled data $U = \{\boldsymbol{x}_j\}_{j=1}^J$ at the server to memorize the ensemble prediction rule, by turning $U$ into a pseudo-labeled set $\mathcal{T}_{pred} = \{(\boldsymbol{x}_j, \boldsymbol{\mu}_j)\}_{j=1}^J$, where $\boldsymbol{\mu}_j = \frac{1}{M}\sum_{m=1}^M p(\boldsymbol{y}|\boldsymbol{x}_j; \boldsymbol{w}^{(m)})$ [6]. Then, we use $\mathcal{T}_{pred}$ as supervision to train $\boldsymbol{w}_{pred}$, aiming to mimic the ensemble prediction rule on $\mathcal{T}_{pred}$. As we want to provide cost-effective UQ, we also propose to send back to the clients a global model $\boldsymbol{w}_{uq}$ (second student model) that mimics the uncertainty rule of the ensemble. We turn $U$ into a pseudo-labeled set $\mathcal{T}_{uq} = \{(\boldsymbol{x}_j, CI_j)\}_{j=1}^J$, where $CI_j = \{\boldsymbol{L}_{CI}^{(j)}, \boldsymbol{U}_{CI}^{(j)}\}$ can be the 95% credible interval for the ensemble of predictions $\{p(\boldsymbol{y}|\boldsymbol{x}_j; \boldsymbol{w}^{(m)})\}_{m=1}^M$ to memorize the ensemble uncertainty rule at the input $\boldsymbol{x}_j$. Then, we use $\mathcal{T}_{uq}$ as supervision to train $\boldsymbol{w}_{uq}$. See the extended justifications related to $\boldsymbol{w}_{uq}$ in Appendix B.2 and B.3. In addition, the algorithm of 2S approach is in Appendix 1, and the diagram can be seen in Figure 2.

## 4   Experiments

We conducted experiments on a classical regression problem [11, 36], to answer two research questions and understand the effectiveness of the 2S. The details of the experiments are in Appendices B.4 and D. This regression problem is a task that allows us to visualize the uncertainty. It consists of

using 2S to enable $\boldsymbol{w}_{pred}$ and $\boldsymbol{w}_{uq}$ to learn (regressing) a cosine function in the interval [-5,9] during 100 communication rounds with clients. Where the train data (server) to build the initial ensemble is corrupted by input-dependent Gaussian noise and limited to the intervals [-4,-3], [-1.62,-0.42], [1.58,2.44],[3.98,4.98], and [6.78,7.38], see Figure 1. The research questions and the respective answers are the following. **Q1)** Can the 2S enable efficient updates of $\boldsymbol{w}_{pred}$ and $\boldsymbol{w}_{uq}$ in the presence of clients with non-i.i.d. data? Yes. Figure 1 shows the predictions done in the server by the ensemble, and in the clients by using $\boldsymbol{w}_{pred}$ and $\boldsymbol{w}_{uq}$. In round 0, their predictions are far from ground truth (red line), which is an expected result [6]. However, after 100 rounds, their predictions converged to the ground truth. It is important to point out that the $\boldsymbol{w}_{pred}$ predictions converged to the ground truth with or without the truncation filter, even though, as expected, the predicted uncertainty via $\boldsymbol{w}_{uq}$ increases without the truncation filter; see the additional results in Appendix E. Furthermore, the $\boldsymbol{w}_{pred}$ predictions demonstrate lower RMSPE values compared to FedAvg, highlighting its superior accuracy; see Table 1 in the Appendix.

**Q2)** Can $\boldsymbol{w}_{uq}$ provide aleatory (uncertainty inherent in the data itself that cannot be reduced) and epistemic (uncertainty due to limited data that can potentially be reduced with more data or better models) uncertainties on the client side? Yes. We compared the $\boldsymbol{w}_{pred}$ and $\boldsymbol{w}_{uq}$ predictions with GP regressions, which can be considered a gold standard for regression uncertainty even though it is less reliable for extrapolation [13, 7, 24]; see Figure 1. If the $\boldsymbol{w}_{pred}$ can capture both types of uncertainty [12], its prediction intervals after 100 rounds should be reduced and include the ground truth values between the interval [-5,9],

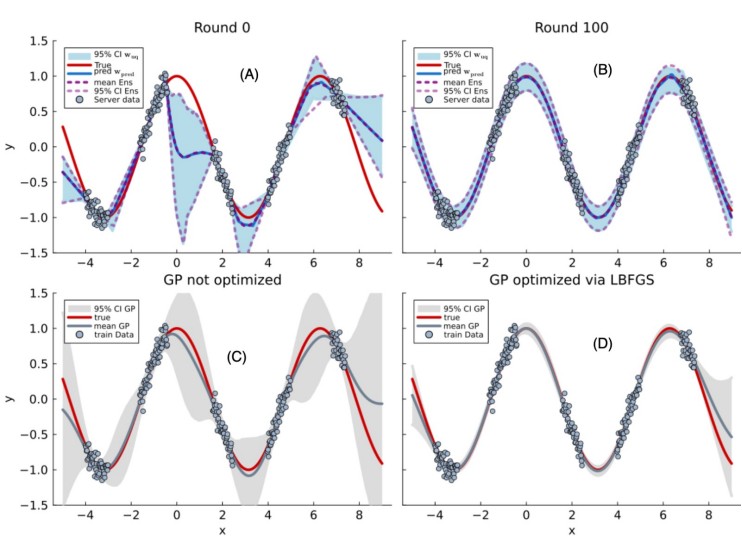

Figure 1: Regression problem. Plots (A) and (B) show the ensemble, $\boldsymbol{w}_{pred}$ and $\boldsymbol{w}_{uq}$ predictions in the rounds 0 and 100. Plots (C) and (D) show the GP predictions before and after optimization.

such as the GP model after optimization. In plot (B), we can see that $\boldsymbol{w}_{pred}$ captures these uncertainties, which is visible as the width of the prediction intervals around the data points where the noise level in the data is consistent. It is important to note that the predicted mean by $\boldsymbol{w}_{pred}$ is closer to the ground truth values than GP model predictions; see the RMSPE values in Figure 4 of the Appendix. This is because the GP model becomes underconfident in the extrapolation zones beyond the observed data range; see Plot (D). The $\boldsymbol{w}_{pred}$ provided more confident extrapolations.

## 5 Discussion

In this work, we proposed the 2S approach to enable predictive UQ in FL clients. The 2S approach distills a BME into two student models: one focused on generating accurate predictions and the other on quantifying uncertainty, while filtering out unreliable client models in FL to ensure robust and efficient updates across decentralized, non-i.i.d. data sources. Our empirical results shown that 2S can effectively manage non-i.i.d. data and provide aleatory and epistemic uncertainty quantification across decentralized clients. **Limitation.** This initial study does not provide a strategy to define an optimum value for the truncation filter threshold $\gamma$ and the best moment to start the truncation filtering. **Future work.** It consists of addressing these limitations beyond demonstrating the applicability and efficiency of 2S in challenging conditions. In real-world applications, particularly in sensitive domains such as healthcare and biomanufacturing, 2S has the potential to facilitate collaborative model development without compromising data privacy. By allowing companies to build robust

models with predictive UQ that can be shared and utilized across different organizations, 2S not only advances the state-of-the-art in FL but also opens new avenues for secure and efficient model training in complex, real-world scenarios.

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

# A    Algorithm of 2S approach

Algorithm 1 and the diagram of Figure 2 present the 2S approach to ensure UQ across decentralized and heterogeneous data sources. The algorithm leverages BME techniques, distilling the ensemble into two compact models (referred to as "students"): one for prediction ($\boldsymbol{w}_{pred}$) and another for uncertainty quantification ($\boldsymbol{w}_{uq}$).

Steps of the Algorithm:

**Step 1 - Server initialization (line 1)**: The server begins with an ensemble of $M$ models, initial versions of the student models $\boldsymbol{w}_{pred}$ and $\boldsymbol{w}_{uq}$, an unlabeled dataset $U = \{x_j\}_{j=1}^{J}$, and an initial ensemble training dataset $\{\bar{\boldsymbol{x}}, \bar{\boldsymbol{y}}\}$.

**Step 2 - Client initialization (line 2)**: Each client $i$ possesses its own labeled local dataset $D_i$.

**Step 3 - Federated training loop (lines 3-17):**

- Sever defines the number of communications rounds R (line 3).
- Client selection (line 4): The server randomly selects a subset $S$ of clients to participate in the training round.
- Model communication (line 5): The server sends the current versions of $\boldsymbol{w}_{pred}$ and $\boldsymbol{w}_{uq}$ to the selected clients.
- Client training (lines 7-8): Each client $i$ in $S$ uses its local dataset $D_i$ to train the predictive model $\boldsymbol{w}_{pred}$, resulting in an updated model $\boldsymbol{w}_i$. These updated models are then sent back to the server.

**Step 4 - Truncation filter for $w_i$ (line 10-16):**

- Prediction (line 11): For each model $\boldsymbol{w}_i$ received from the clients, the server generates predictions $\{\boldsymbol{y}_q'\}_{q=1}^{Q}$ for the test dataset $\bar{\boldsymbol{x}}$.
- Credible interval calculation (line 12): The server calculates the credible intervals $\{[\boldsymbol{L}_{CI}^{(q)}, \boldsymbol{U}_{CI}^{(q)}]\}_{q=1}^{Q}$ using either the ensemble predictions. Furthermore, we used a 95% credible interval, but a confident interval can be used too.
- Drift detection (lines 13-15): The server calculates the percentage $\delta$ of predictions $\{\boldsymbol{y}_q'\}_{q=1}^{Q}$ that fall within the credible intervals $\{[\boldsymbol{L}_{CI}^{(q)}, \boldsymbol{U}_{CI}^{(q)}]\}_{q=1}^{Q}$. If $\delta$ is below a threshold $\gamma$, the model $\boldsymbol{w}_i$ is considered drifted and is (not included or) excluded from the ensemble.

**Step 5 - Knowledge distillation (lines 18-22):**

- New ensemble construction (line 18): The server updates the ensemble by combining the filtered client models $\{\boldsymbol{w}_i; i \in S\}$ with the previous ensemble models $\{\boldsymbol{w}^{(m)}\}_{m=1}^{M}$.
- Pseudo-labeled data construction (lines 19-20): The server constructs pseudo-labeled datasets $\mathcal{T}_{pred}$ and $\mathcal{T}_{uq}$ using the ensemble's predictions on the unlabeled dataset $U$.
- Knowledge distillation (lines 21 - 22): The server updates the student models $\boldsymbol{w}_{pred}$ and $\boldsymbol{w}_{uq}$ by training them on the pseudo-labeled datasets $\mathcal{T}_{pred}$ and $\mathcal{T}_{uq}$, respectively. The knowledge distillation can be implemented in different ways. In our experiments, we use an MSE loss with Adam optimization. However, we can use a cross-entropy loss for the classification task and perform the minimization with SGD or SWA [20, 6].

**Step 6 - Server output (line 24):** After completing the final round R, the server outputs the updated models $\boldsymbol{w}_{pred}$ and $\boldsymbol{w}_{uq}$

**Algorithm 1** 2S (2 Students)

---

1: **Server input:** Ensemble of M models, initial students $\boldsymbol{w}_{pred}$ and $\boldsymbol{w}_{uq}$, unlabeled data $U = \{\boldsymbol{x}_j\}_{j=1}^J$, ensemble initial training data $\{\bar{\boldsymbol{x}} = \{\bar{\boldsymbol{x}}_q\}_{q=1}^Q, \bar{\boldsymbol{y}} = \{\bar{\boldsymbol{y}}_q\}_{q=1}^Q\}$

2: **Client $i$ input:** local labeled data $D_i$

3: **for** $r = 1$ to $R$ **do**

4:      Sample clients $S \subseteq \{1, \ldots, N\}$

5:      Communicate $\boldsymbol{w}_{pred}$ and $\boldsymbol{w}_{uq}$ to all clients $i \in S$

6:      **for** each client $i \in S$ **in parallel do**

7:          $\boldsymbol{w}_i \leftarrow \text{ClientTraining}(\boldsymbol{w}_{pred}, D_i)$

8:          Communicate $\boldsymbol{w}_i$ to the server

9:      **end for**

10:      **for** each trained model from client $i \in S$ **in parallel do**

11:          Predict $\{\boldsymbol{y}'_q\}_{q=1}^Q = p(\bar{\boldsymbol{y}}|\bar{\boldsymbol{x}}, \boldsymbol{w}_i)$

12:          Predict $\{[\boldsymbol{L}_{CI}^{(q)}, \boldsymbol{U}_{CI}^{(q)}]\}_{q=1}^Q$ # using predictions from ensemble or global model $f_{uq}(\bar{\boldsymbol{x}}, \boldsymbol{w}_{uq})$

13:          Calculate the percentage $\delta$ of $\{\boldsymbol{y}'_q\}_{q=1}^Q$ within $\{[\boldsymbol{L}_{CI}^{(q)}, \boldsymbol{U}_{CI}^{(q)}]\}_{q=1}^Q$

14:          **if** $\delta < \gamma$ **then**

15:             remove $\boldsymbol{w}_i$ from set $\{\boldsymbol{w}_i; i \in S\}$ # remove drifted models

16:          **end if**

17:      **end for**

18:      Construct the new ensemble $\{\boldsymbol{w}^{(m')}\}_{m'=1}^{M'} = \{\boldsymbol{w}_i; i \in S\} \cup \{\boldsymbol{w}^{(m)}\}_{m=1}^M$

19:      Construct $\mathcal{T}_{pred} = \{(\boldsymbol{x}_j, \boldsymbol{\mu}_j)\}_{j=1}^J$, where $\boldsymbol{\mu}_j = \frac{1}{M} \sum_{m'}^{M'} p(\boldsymbol{y}|\boldsymbol{x}_j; \boldsymbol{w}^{(m')})$

20:      Construct $\mathcal{T}_{uq} = \{(\boldsymbol{x}_j, CI_j)\}_{j=1}^J$, where $CI_j$ can be the 95% of credible interval

21:      $\boldsymbol{w}_{pred} \leftarrow \text{KnowledgeDistillation}(\mathcal{T}_{pred}, \boldsymbol{w}_{pred})$

22:      $\boldsymbol{w}_{uq} \leftarrow \text{KnowledgeDistillation}(\mathcal{T}_{uq}, \boldsymbol{w}_{uq})$

23: **end for**

24: **Server output:** $\boldsymbol{w}_{pred}$ and $\boldsymbol{w}_{uq}$

---

# B  Extented Justifications

## B.1  Proposed method to generate $q(\boldsymbol{w})$

To justify using the trained models from clients as a method to generate the approximate posterior distribution $q(\boldsymbol{w})$ in the context of Federated Learning (FL), we can consider several points:

**Implicit posterior sampling (sampling without additional computation)[29]:** By leveraging the models already trained by clients during the federated learning process, we obtain samples from an implicit distribution that approximates the true posterior. This avoids the need for additional sampling procedures, which would otherwise require extra communication rounds or increased computational costs. Since these models are already trained under different data, they represent samples from a distribution that approximates the true posterior $p(\boldsymbol{w}|D)$.

**Efficient use of available information (utilizing existing models)[39]:** The use of trained models from clients makes efficient use of the existing information in the FL setting. Rather than discarding the diversity in model parameters learned by individual clients, these models can be aggregated into an ensemble that represents an approximation of the posterior distribution. This approach is practical because it aligns with the natural workflow of FL, where multiple models are already being trained and updated iteratively across different clients.

**Approximation of uncertainty (credible intervals from local models)[10]:** The 95% credible intervals (CIs) calculated from the ensemble of local models provide a measure of uncertainty in the aggregated model's predictions. Since each local model contributes to the overall uncertainty by reflecting its unique view of the data distribution, using these models as samples for $q(\boldsymbol{w})$ naturally incorporates the variance arising from different client data. This is particularly useful in FL, where the goal is to develop a global model that is robust to variations across clients.

**Theoretical basis from ensemble methods (connection to ensemble learning theory)[43]:** Ensemble learning methods, such as bagging and boosting, rely on combining multiple models trained on

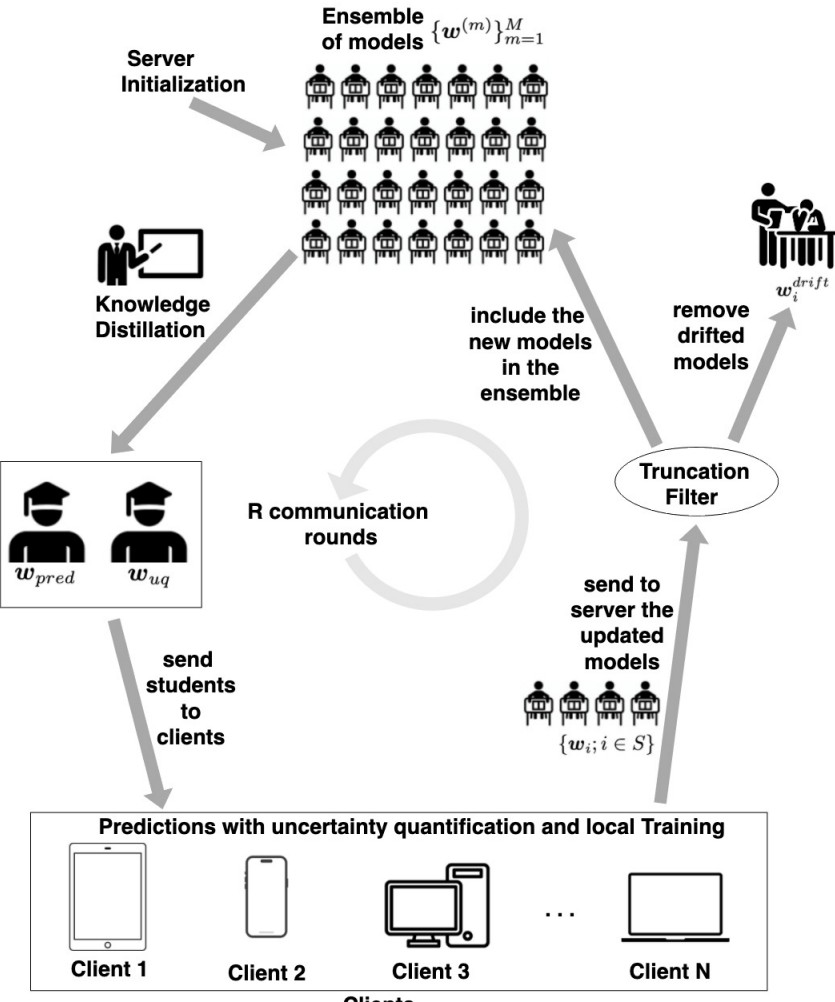

Figure 2: Overview of the 2S (two student) approach in federated learning. The process begins with server initialization, where an ensemble of models is created and maintained. Client models undergo a truncation filter to remove drifted models before updating the ensemble. Knowledge distillation is then applied to compress the ensemble into two compact student models: one for predictions ($w_{pred}$) and another for uncertainty quantification ($w_{uq}$). These student models are sent to clients for local training and prediction, with the process iteratively refined over multiple communication rounds.

different subsets or versions of the data to achieve a better approximation of the true model distribution. In FL, each client's model can be considered analogous to a base learner trained on a specific subset of the global data (the client's local data). Therefore, combining these client models aligns with ensemble learning principles, where the goal is to approximate the underlying data distribution more accurately.

**Scalable and adaptable approach (scalability)[19]:** Using trained models from clients scales naturally with the FL framework. As more clients participate or as more communication rounds are performed, the ensemble grows richer, potentially providing a more accurate approximation of the posterior distribution.

**Adaptability to new data:** Since the client models are continually updated based on new local data in each round, the ensemble dynamically adapts to changes in data distribution, making the approximation $q(\boldsymbol{w})$ more responsive to new information.

## B.2 Global Model $\boldsymbol{w}_{uq}$ to Mimics the Ensemble Uncertainty Rule

Sending back a global model $\boldsymbol{w}_{uq}$ that mimics the ensemble uncertainty rule is a novel approach in FL, particularly for enabling "cost-effective" UQ on client devices. Here are three justifications for this approach:

**Practicality in FL:** Limited resources on clients. Client devices in FL are often resource-constrained (e.g., smartphones, IoT devices). Training complex models or ensembles directly on these devices may be impractical. By sending back a distilled model $\boldsymbol{w}_{uq}$, which has learned to mimic the uncertainty characteristics of the ensemble, the need for heavy computation on the client side is reduced, making UQ feasible with limited computational resources.

**Consistency with Bayesian principles:** i) Approximation of posterior distribution: In a Bayesian context, uncertainty quantification is about approximating the posterior distribution of the model parameters given the data. The ensemble represents an approximation of this posterior, and by mimicking the ensemble's uncertainty rule, $\boldsymbol{w}_{uq}$ effectively approximates the posterior distribution in a way that is computationally tractable for client devices. ii) Update of uncertainty: by learning from the ensemble's credible intervals, $\boldsymbol{w}_{uq}$ ensures that the uncertainty estimates on the client side are calibrated with respect to the global data distribution. This is crucial for making reliable predictions, particularly in scenarios where data distributions across clients vary.

**Addressing model drift[14]:** The inclusion of $\boldsymbol{w}_{uq}$, which captures the ensemble's uncertainty rule, can help in identifying and mitigating model drift on the client side. Clients can use the uncertainty estimates provided by $\boldsymbol{w}_{uq}$ to assess whether their local model is drifting from the global model, allowing for more informed updates and reducing the risk of propagating drifted models back to the server.

## B.3 Proposed method to create $\boldsymbol{w}_{uq}$

The method to create $\boldsymbol{w}_{uq}$, which is a global model (student model) that mimics the ensemble uncertainty rule, aligns with several well-established principles in machine learning and uncertainty quantification. Below are justifications for the strategy:

**Knowledge distillation for uncertainty quantification:** i) The concept of knowledge distillation is well-established in the machine learning literature, where a "teacher" model (usually more complex or an ensemble of models) transfers its knowledge to a simpler "student" model. This technique has been successfully applied in various contexts to reduce the model's complexity while retaining most of the predictive power of the teacher model [20, 34]. In this case, the teacher model is the Bayesian ensemble, which inherently captures uncertainty. Then, a student model $\boldsymbol{w}_{pred}$ learns the predictive mean, and another one $\boldsymbol{w}_{uq}$ learns the uncertainty bounds (credible intervals), enabling them to approximate the moments of the ensemble. ii) Memorizing Uncertainty: By turning the unlabeled dataset $U$ into a pseudo-labeled set $\mathcal{T}_{uq} = \{(\boldsymbol{x}_j, CI_j)\}_{j=1}^{J}$, where each $CI_j$ is the credible interval from the ensemble, we can encode the ensemble's uncertainty into the student model. This allows $\boldsymbol{w}_{uq}$ to learn the variability or confidence in ensemble predictions, which is crucial for uncertainty quantification.

**Use of credible intervals (CIs):** i) Representing uncertainty: CIs are a common Bayesian approach to quantify uncertainty. The use of CIs in $\mathcal{T}_{uq}$ as labels for training $\boldsymbol{w}_{uq}$ is appropriate because CIs directly represent the range within which the true predictions are expected to lie with a certain probability (e.g., 95%). This ensures that $\boldsymbol{w}_{uq}$ is trained to understand and replicate the uncertainty characteristics of the ensemble, which is the essence of the uncertainty rule. ii) Calibration of Predictions: Training $\boldsymbol{w}_{uq}$ using CIs can help in calibrating the model's predictions, making them more reliable in practice. Proper calibration is critical in applications where decision-making depends on the model's confidence in its predictions.

**Supervision using pseudo-labels:** i) Learning from unlabeled data: The use of unlabeled data $U$ to create pseudo-labels $\mathcal{T}_{uq}$ is important, especially in scenarios where labeled data is scarce or expensive to obtain [6]. By leveraging the ensemble's predictions to generate pseudo-labels, the model can be trained without the need for additional labeled data, making the process more efficient and scalable. ii) Consistency across models: Training $\boldsymbol{w}_{uq}$ on the pseudo-labeled set $\mathcal{T}_{uq}$ ensures that the uncertainty estimates it learns are consistent with those of the ensemble, maintaining coherence across different models in the FL framework.

## B.4 Synthetic Dataset mimicking health-care and biomanufactruing data

The regression problem used in the experiment, which involves learning a noisy cosine function, serves as a simplified yet effective representation of challenges encountered in healthcare data modeling [1, 3, 30]. Therefore, it can be used as an ideal testbed for mimicking the complexities of healthcare data. This was done to assess if the 2S approach can offer a practical solution for deploying robust ML models in healthcare devices through federated learning, ensuring that both predictive accuracy and uncertainty are appropriately managed across decentralized and privacy-sensitive environments.

In healthcare, models often need to learn from data that is both noisy and subject to variability across different patient groups or sensor readings, akin to the input-dependent Gaussian noise used in the regression task [35]. The task of regressing the cosine function under such conditions mimics the challenges faced by models in predicting physiological signals or patient outcomes, where both signal noise and data sparsity can lead to significant uncertainties in predictions [8].

The 2S approach, which involves distilling the knowledge from a BME into two compact student models, is particularly advantageous in healthcare settings. In federated learning (FL) for healthcare, data is often distributed across multiple institutions or devices, with privacy concerns preventing centralized data aggregation. The 2S approach allows for efficient and "cost-effective" uncertainty quantification (UQ) at the client side by creating models that are not only lightweight but also capable of capturing both aleatoric and epistemic uncertainties. By leveraging the 2S approach in FL, healthcare devices can improve their predictive accuracy and reliability. The student model focused on UQ ($\boldsymbol{w}_{uq}$) can provide clinicians or healthcare devices with credible intervals that reflect the confidence in the model's predictions, which is essential for risk assessment and decision-making in clinical settings. Meanwhile, the other student model ($\boldsymbol{w}_{pred}$) ensures that the predictions are as accurate as possible, adapting to new data as it becomes available through FL.

Healthcare data often suffers from heterogeneity and non-i.i.d. (non-independent and identically distributed) characteristics across different institutions or patient groups, as highlighted in the literature [1, 35, 8, 3, 30]. Therefore, we assess if the 2S approach can effectively mitigate these issues by filtering out models that do not align with the broader data distribution, ensuring that only high-quality models contribute to the global model. This assessment is important to check if 2S can be used to enhance the robustness of FL in healthcare applications, where model drift due to non-i.i.d. data is a significant concern.

It is important to point out that the 2S approach could also be used to improve ML model training in biopharmaceutical companies [31, 16]. The 2S approach presents an opportunity for advancing the monitoring and optimization of complex biomanufacturing processes, such as adeno-associated virus (AAV) production. In this highly competitive and regulated industry, data sharing between companies is limited due to the proprietary nature and high cost of the data, making it challenging to train robust ML models that can generalize across different manufacturing setups [31, 16]. However, by leveraging a consortium of biopharmaceutical companies, the 2S approach can enable the collaborative development of powerful ML models without requiring the direct exchange of sensitive data. In this setup, an initial global model can be trained and then shared among the companies, each acting as

a client in a federated learning framework. Through the 2S approach, this model can be iteratively refined and updated as it is exposed to data from different companies' bioprocesses. The two-student (2S) strategy allows the model to learn both the predictive mean and uncertainty quantification from each client's data, enhancing the model's robustness and reliability. This process results in a final ML model that accurately captures the intricacies of bioprocesses, such as AAV production, and can be confidently deployed across the industry. By using 2S, biopharmaceutical companies can collaboratively build and benefit from advanced ML models while safeguarding their proprietary data, ultimately driving innovation and efficiency in biomanufacturing.

## C   Related Work

The literature on uncertainty quantification (UQ) in federated learning (FL) is relatively sparse compared to the broader field of UQ in deep learning [36]. However, we hereby discuss the following two lines of work that are closely related to the 2S approach.

**Handling non-i.i.d. data in FL.** Traditional FL methods like FedAvg [27] and its variants assume that all client models contribute equally, which can result in suboptimal performance when client data distributions are highly heterogeneous (non-i.i.d.). Recent approaches such as FedProx [21] and personalized FL methods [37] introduce regularization or personalization techniques to address non-i.i.d. data. However, these methods typically do not leverage UQ as a criterion for model selection. BME approaches, like FedBE [6] and FedPPD [4], implement Bayesian inference on the server, effectively utilizing client information to mitigate performance degradation, particularly for local models trained on non-i.i.d. data. The 2S approach advances this by directly addressing the non-i.i.d. challenge through the use of credible intervals derived from ensemble predictions. These intervals help to identify models that are likely trained on data distributions closely aligned with the global distribution, adding a novel dimension to model selection and aggregation in FL. Moreover, while BME approaches are grounded in Bayesian principles, they do not provide predictive UQ, a gap that the 2S approach fills. It is important to point out that conformal prediction [38] was designed recently for federated context [33, 23]. Federated Conformal Prediction (FCP) [23] is an extension of Conformal Prediction, where multiple clients potentially have different local data distributions. Traditional conformal prediction assumes the exchangeability of data, but in a federated environment, this assumption is commonly broken due to heterogeneity across clients. Although the FCP provides marginal coverage guarantees for the prediction sets on unseen data from the global distribution, the 2S approach can handle drift models with the truncation filter, reducing the errors caused by noisy data. In addition, for not modeling uncertainty about the model parameters or structure, FCP does not provide epistemic uncertainty, while 2S provides.

**Uncertainty quantification in FL.** Bayesian Neural Networks (BNNs) have been used in FL to enable local uncertainty quantification while using deep neural networks (DNNs) for task learning, thereby enhancing FL robustness and performance, especially on limited data [4, 5, 41]. However, BNNs present significant challenges in FL, such as high computational and memory costs for training local models, particularly when the model parameter scale is large. Additionally, selecting appropriate prior distributions for local model parameters can be difficult, especially when complex relationships between model outputs and parameters need to be estimated. The 2S approach overcomes these challenges by distilling two global models that mimic the predictive mean and uncertainty rules of the ensemble. While ensemble distillation of mean and variance is not entirely new [26], it has not been applied in this manner within the FL context. The 2S approach uses unlabeled data to train $\boldsymbol{w}_{uq}$ on the server side, avoiding additional training overhead on the client side. This approach aligns with the need for efficiency in FL environments, where communication and computational resources are often limited. By computing ensemble predictions once on the server and using them to train $\boldsymbol{w}_{uq}$, the 2S approach eliminates the need for repeated ensemble calculations on the client side, conserving computational resources and reducing latency. This makes sending a distilled model like $\boldsymbol{w}_{uq}$ to clients a practical solution, ensuring that clients can perform UQ without the burden of handling the full ensemble.

## D Experiment details

In the experiment, the truncation filter threshold $\gamma$ used was 80%, and the truncation filtering started at round 2. However, this initial study does not provide a strategy to define an optimum value for the truncation filter threshold $\gamma$ and the best moment to start the truncation filtering.

### D.1 Task

The task of the experiment has stages in the server and in clients. The task consists of regressing a cosine function in the interval [-5,9] during 100 communication rounds with clients via 2S approach. The task enables us to visualize the aleatoric uncertainty (representing the uncertainty inherent in the data itself), and epistemic uncertainty quantification (reflecting the uncertainty due to limited data or model complexity) in the prediction of $\boldsymbol{w}_{pred}$ and $\boldsymbol{w}_{uq}$ .

#### D.1.1 Stage in server

The initial training data $D_{itd} = \{\bar{\boldsymbol{x}} = \{\bar{\boldsymbol{x}}_q\}_{q=1}^{Q}, \bar{\boldsymbol{y}} = \{\bar{\boldsymbol{y}}_q\}_{q=1}^{Q}\}$ (in the server) to build the initial ensemble is corrupted by input-dependent Gaussian noise, by following

$$y(x) = \cos(x) + \epsilon(x), \quad \epsilon(x) \sim \mathcal{N}(0, 0.1). \tag{1}$$

The trained data is limited to the intervals [-4,-3], [-1.62,-0.42], [1.58,2.44],[3.98,4.98], and [6.78,7.38], and have the size 237 data points, see Figure 1. Furthermore, $D_{itd}$ is the same data used in the filtering step of the 2S approach to avoid drifted models (see lines 10-15 of algorithm 1).

The initial ensemble consists of 50 submodels, which are neural networks. The neural network (NN) model $h(\boldsymbol{x}, \boldsymbol{w})$ used in the task consists of an input layer with one neuron, two hidden layers each with 20 neurons using ReLU activation functions, and a single-neuron output layer with a linear activation. It has a total of 481 parameters (including weights and biases). This neural network architecture is the same one used to build the $\boldsymbol{w}_{pred}$ (student 1), $h_{pred}(\boldsymbol{x}, \boldsymbol{w}_{pre})$. The $\boldsymbol{w}_{uq}$ (student 2) has to predict the 2.5th percentile (output1) and the 97.5th percentile (output2) of the ensemble set of predictions that represents the 95% credible interval. Therefore, a similar NN architecture was used for $h_{uq}(\boldsymbol{x}, \boldsymbol{w}_{uq})$. The difference is that $h_{uq}(\boldsymbol{x}, \boldsymbol{w}_{uq})$ has a two-neuron output layer and a total of 502 parameters (including weights and biases).

The optimizations of $h_{pred}$ and $h_{uq}$ during the knowledge distillation are performed with a mean squared error (MSE) loss, with $\mathcal{T}_{pred}$ and $\mathcal{T}_{uq}$, through the ADAM algorithm. $\mathcal{T}_{pred}$ and $\mathcal{T}_{uq}$ are composed of unlabeled data $U = \{\boldsymbol{x}_j\}_{j=1}^{J=701}$ of 701 data points.

#### D.1.2 Stage in clients

The experiment simulates a total of 135 clients, each one with a unique labeled dataset $D_i$ with only 50 data points to update (training) the $\boldsymbol{w}_{pred}$ and $\boldsymbol{w}_{uq}$. 100 clients with only i.i.d data and 35 clients have $D_i$ composed of non-i.i.d. data.

**100 clients with i.i.d. data.** 100 clients have unique $D_i$ with 50 data points obtained from:

$$D_i = \{(x_{ij}, y_{ij}) \mid x_{ij} \sim \mathcal{U}(-5, 9), y_{ij} = \cos(x_{ij}) + \epsilon_{ij}, \epsilon_{ij} \sim \mathcal{N}(0, 0.1)\}_{j=1}^{50} \tag{2}$$

The input data $x_i$ for client $i$ is sampled uniformly from the interval $[-5, 9]$, $x_i \sim \mathcal{U}(-5, 9)$. The cosine function is corrupted by Gaussian noise, and each client $i$ receives a dataset $D_i$ consisting of 50 data points $(x_{ij}, y_{ij})$ where $j = 1, 2, \ldots, 50$.

**5 clients with non-i.i.d. data of a different data range.** For each of the 5 clients, the data range is limited to the interval $[-5, 2]$, and the cosine function is corrupted by Gaussian noise as follows,

$$D_i = \{(x_{ij}, y_{ij}) \mid x_{ij} \sim \mathcal{U}(-5, 2), \ y_{ij} = \cos(x_{ij}) + \epsilon_{ij}, \ \epsilon_{ij} \sim \mathcal{N}(0, 0.1)\}_{j=1}^{50} \quad \text{for } i = 1, 2, \ldots, 5 \tag{3}$$

**5 clients with non-i.i.d. data related to a different data range.**

For each of the 5 clients, the input data is sampled from $[-5, 9]$, but the noise level is scaled differently (0.5 times a standard normal distribution) as follows,

$$
\begin{aligned}
D_i = \{(x_{ij}, y_{ij}) \mid x_{ij} \sim \mathcal{U}(-5, 9), \ y_{ij} = \cos(x_{ij}) + \epsilon_{ij} + \gamma_{ij}, \\
\epsilon_{ij} \sim \mathcal{N}(0, 0.1), \\
\gamma_{ij} \sim \mathcal{N}(0, 1) \times 0.5\}_{j=1}^{50} \\
\text{for } i = 1, 2, \dots, 5.
\end{aligned}
\tag{4}
$$

**25 clients with non-i.i.d. data related to a mixed non-i.i.d. data generation.**

For each of the 5 clients, the input data is sampled from the interval $[-5, 2]$ with a small Gaussian noise and a bias term added to the cosine function as follows,

$$
\begin{aligned}
D_i = \{(x_{ij}, y_{ij}) \mid x_{ij} \sim \mathcal{U}(-5, 9), \ y_{ij} = \cos(x_{ij}) + \epsilon_{ij} + \gamma_{ij} + 0.15, \\
\epsilon_{ij} \sim \mathcal{N}(0, 0.1), \\
\gamma_{ij} \sim \mathcal{N}(0, 1) \times 0.1\}_{j=1}^{50} \\
\text{for } i = 1, 2, \dots, 25.
\end{aligned}
\tag{5}
$$

## D.2 Baselines

### D.2.1 FedAvg

FedAvg is adopted as a baseline. All the clients in each communication round are trained with the same amount of data ( $D_i$ with only 50 data points), such as for the 2S approach. Therefore, all client models will have the same weight at the aggregation step. Since it is the standard aggregation method it is evaluated in scenarios with only i.i.d. data (total of 100 clients) and with a mix of non-i.i.d. data and i.i.d. data (total of 135 clients). It is important to point out that we report mean ± standard deviation (std) over 10 times of experiments for 2S and FedAvg, see Table 1.

### D.2.2 Gaussian process model

In the experiment, a Gaussian Process (GP) model is employed to regress the cosine function, where the training data is corrupted by Gaussian noise. The GP provides a non-parametric Bayesian approach to regression, offering a principled way to quantify uncertainty in predictions [13, 7, 24]. Therefore, it is used as our baseline.

The GP was implemented with GaussianProcesses.jl and the specific components of the GP model used in this experiment are as follows:

- Mean Function $m(x)$, which represents the expected value of the GP at any input $x$. In this case, a zero mean function is assumed, i.e., $m(x) = 0$. This choice reflects a prior belief that, in the absence of data, the function being modeled is centered around zero. This is implemented in the code using the 'MeanZero()' function.

- Kernel (Covariance Function) $k(x, x')$ that defines the covariance between pairs of inputs $x$ and $x'$. It encodes assumptions about the smoothness, periodicity, and other properties of the function being modeled. In this experiment, the Squared Exponential (SE) kernel is utilized, which is a common choice for modeling smooth functions. The SE kernel is defined as:

$$
k(x, x') = \sigma_f^2 \exp\left(-\frac{(x - x')^2}{2\ell^2}\right)
$$

where $\sigma_f^2$ is the signal variance, controlling the vertical variation of the function, and $\ell$ is the length scale, controlling the smoothness of the function. In the implementation, the parameters of the kernel are initially set to 0.0 on the log scale ('SE(0.0, 0.0)'), allowing the model to learn appropriate values during training.

- Observation noise accounts for the variability in the data that cannot be explained by the underlying function alone. The noise is modeled as Gaussian with zero mean and variance $\sigma_n^2$. In this experiment, the logarithm of the standard deviation of the observation noise is set to -1.0, implying $\sigma_n = \exp(-1.0)$. This parameter controls the amount of noise the model expects in the observations.

- The training data ($D_{itd}$) used in GP model is the same one used to train the initial ensemble. See Figure 1.

- Gaussian Process Prior. Before observing any data, the GP is specified by a prior distribution over functions. Given the zero mean function and the SE kernel, the GP prior is $f(x) \sim \mathcal{GP}(0, k(x, x'))$. This prior reflects the assumption that the function is smooth (as determined by the kernel) and centered around zero.

- Posterior Inference. After observing the data $D_{te}$, the GP updates its beliefs, resulting in a posterior distribution over functions. The posterior GP is then used to make predictions at new input points $x^*$. The posterior predictive distribution at a new input $x^*$ is Gaussian with mean $\mu(x^*)$ and variance $\sigma^2(x^*)$, given by:

$$\mu(x^*) = k(x^*, X)[K(X, X) + \sigma_n^2 I]^{-1} y$$
$$\sigma^2(x^*) = k(x^*, x^*) - k(x^*, X)[K(X, X) + \sigma_n^2 I]^{-1} k(X, x^*)$$

where $X$ is the matrix of training inputs, $y$ is the vector of training outputs, and $k(x^*, X)$ is the covariance vector between the new input $x^*$ and the training inputs. This process enables the GP to provide predictions with associated uncertainty, reflecting both the underlying function and the noise in the data.

- Optimization of the GP Model. To improve the model's performance, the GP hyperparameters (such as the kernel parameters and observation noise) are optimized using the 'LBFGS' method, a popular quasi-Newton optimization algorithm. See Figure 1.

### D.3 Metrics

A test set of 1000 data points was used to assess the performance (extrapolations) of the GP model (before and after optimization), the $\boldsymbol{w}_{pred}$, and $\boldsymbol{w}_{uq}$ over the communication rounds; see Figure 1, 3 and 4. The metrics used are root mean square percentage error (RMSPE) and average interval width analysis (AIWA).

The RMSPE is a metric used to evaluate the accuracy of FedAvg, GP model, $\boldsymbol{w}_{pred}$ and $\boldsymbol{w}_{uq}$ predictions relative to the actual ground truth values of the cosine function, see Figure 4.

AIWA involves calculating the average width of confidence intervals (CIs) produced by two different approaches across a set of observations. The focus is on understanding the relative uncertainty captured by each method, with the mean interval width serving as a proxy for the level of uncertainty. The approach with the larger mean width provides, on average, wider confidence intervals. Wider CIs typically indicate more uncertainty in the predictions, whereas narrower CIs suggest more precise estimates. To quantitatively compare the confidence intervals (CIs) generated by two approaches, A (GP model) and B ($\boldsymbol{w}_{uq}$), we calculate the width of each CI for every observation $i$ as $\delta_A^{(i)} = \text{ub}_A^{(i)} - \text{lb}_A^{(i)}$ for approach A and $\delta_B^{(i)} = \text{ub}_B^{(i)} - \text{lb}_B^{(i)}$ for approach B, where $\text{lb}_A^{(i)}$ and $\text{ub}_A^{(i)}$ are the lower and upper bounds from approach A, and $\text{lb}_B^{(i)}$ and $\text{ub}_B^{(i)}$ are from approach B. We then compute the mean width of the CIs across all $n$ observations as $\bar{\delta}_A = \frac{1}{n} \sum_{i=1}^n w_A^{(i)}$ and $\bar{\delta}_B = \frac{1}{n} \sum_{i=1}^n w_B^{(i)}$. Comparing these mean widths, $\bar{\delta}_A$ and $\bar{\delta}_B$, allows us to determine which approach produces wider CIs on average, where $\bar{\delta}_A > \bar{\delta}_B$ indicates that approach A has wider intervals, $\bar{\delta}_A < \bar{\delta}_B$ indicates approach B has wider intervals, and $\bar{\delta}_A = \bar{\delta}_B$ suggests both approaches yield similar uncertainty levels, see Figure 4.

### D.4 Computing infrastructure used for running experiments

The experiments were conducted using python and Julia programming language (version 1.8.5) on a computing infrastructure with an Intel® Core™ i9-9900K CPU, 62.7 GiB of RAM, and an NVIDIA GeForce RTX 2080 Ti GPU, running Ubuntu 20.04.5 LTS. The experiments with FedAvg were conducted only in Python.

Table 1: Mean $\pm$ standard deviation (std) of RMSPE for 2S and FedAvg with i.i.d. and non-i.i.d. datas.

| Round | 2S (mean $\pm$ std) | FedAvg (mean $\pm$ std) | FedAvg (mean $\pm$ std) |
|---|---|---|---|
| 0 | $7.258 \pm 0.006$ | $13.869 \pm 0.706$ | $13.912 \pm 0.478$ |
| 1 | $3.008 \pm 0.569$ | $8.458 \pm 1.025$ | $9.604 \pm 0.590$ |
| 25 | $0.469 \pm 0.109$ | $0.645 \pm 0.175$ | $0.833 \pm 0.227$ |
| 50 | $0.365 \pm 0.062$ | $0.621 \pm 0.165$ | $0.936 \pm 0.238$ |
| 75 | $0.329 \pm 0.029$ | $0.633 \pm 0.159$ | $0.859 \pm 0.250$ |
| 100 | $0.329 \pm 0.023$ | $0.625 \pm 0.166$ | $0.931 \pm 0.460$ |
| Client data | i.i.d. and non-i.i.d. | i.i.d | i.i.d. and non-i.i.d. |

# E  Additional Results

Table 1 highlights the superior performance of the 2S approach compared to FedAvg in terms of RMSPE across various training rounds under both i.i.d. and non-i.i.d. data distributions. Initially, at Round 0, 2S starts with a significantly lower RMSPE of 7.258, outperforming both FedAvg implementations, which show RMSPEs around 13.9. As training progresses, 2S continues to reduce its error rates more effectively than FedAvg, achieving an RMSPE of 0.365 by Round 50, and further stabilizing at 0.329 in the later rounds (75 and 100), with minimal standard deviation, indicating consistent and reliable performance. In contrast, FedAvg, while improving, does not match the accuracy or consistency of 2S, particularly in scenarios involving non-i.i.d. data, where the gap between the methods becomes more pronounced. These results demonstrate the robustness and efficiency of the 2S approach in handling data heterogeneity and delivering more accurate predictions in federated learning environments.

As reported in the literature, the ensemble prediction can be noisy in the early rounds of FL [6], affecting the updates of $w_{pred}$, and $w_{uq}$ via knowledge distillation. However, 2S overcomes this issue and enables proper updates of $w_{pred}$, and $w_{uq}$ from an initial poor ensemble, see Figures 1. Furthermore, the appropriate update of $w_{pred}$ occurred even without the truncation filter, even though it increases the predicted uncertainty via $w_{uq}$, see Figure 3. The predicted uncertainty via $w_{uq}$ represents the 2.5th percentile (output1) and the 97.5th percentile (output2) of 95% CIs, and it increases from round 0 to 100, see Figure 3. It is important to note that those results are confirmed with RMSPE and AIWA computed over the communication rounds, see Figure 4. Furthermore, the prediction of $w_{pred}$ (with or without the truncation filter) presented lower RMSPE values than GP model one, see Plots (A) and (B). The AIWA values related to $w_{uq}$ with truncation filter converged to a value close to the AIWA value obtained by GP model optimized via LBFGS while $w_{uq}$ without truncation filter diverged, see Plots (C) and (D). In the experiments, the truncation filter started after 2 rounds. Therefore, we can see that after the 2 rounds the AIWA values started to decrease, see Plot (C) in Figure 4.

Figure 3: Predictions of the ensemble, $w_{pred}$, and $w_{uq}$ over the communication rounds. The round 0 represents the prediction based on the initial ensemble. The initial predictions do not include the ground truth value. However, 2S updates the ensemble and enables the convergence to the ground truth value. In addition, without the truncation filter, the predicted uncertainty by ensemble and $w_{uq}$ increase, but the predicted mean by ensemble and $w_{pred}$ converge to the ground truth.

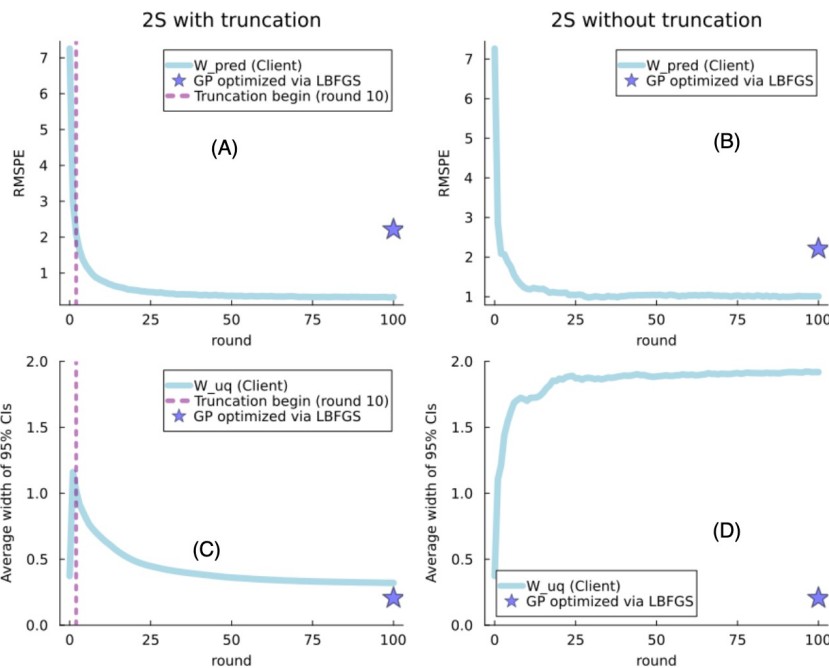

Figure 4: Comparison of Root Mean Square Percentage Error (RMSPE) and Average Interval Width Analysis (AIWA) over Communication Rounds. Subplots (A) and (B) present the RMSPE values, where the 2S approach (with and without the truncation filter) is compared against a Gaussian Process (GP) model. Subplots (C) and (D) display AIWA values, highlighting the differences in uncertainty estimation between the models.

