# OpenReview forum: "Two Students: Enabling Uncertainty Quantification in Federated Learning Clients"
_NeurIPS.cc/2024/Workshop/BDU — NeurIPS BDU Workshop 2024 Poster_

### Official Review · Reviewer_ZWUk · 2024-09-23
**Two Students Approach for Uncertainty Quantification in Federated Learning: An Innovative but Incomplete Solution**

**Rating:** 5
**Confidence:** 4

**Review:**

Summary:
This study proposes the "Two Students" (2S) approach to address uncertainty quantification (UQ) in federated learning (FL). The method involves distilling a Bayesian model ensemble (BME) into two student models: one for accurate predictions and another for UQ, using credible intervals to filter unreliable models. The approach is validated through regression tasks, demonstrating its scalability and effectiveness.

Strengths:
1. Innovative approach combining ensemble learning with uncertainty quantification in federated learning.
2. Novel use of credible intervals to filter drifted models, enhancing robustness in non-i.i.d. data environments.
3. Efficient and scalable solution that enables cost-effective UQ on client devices with minimal computational overhead.

Weaknesses:
1. The study lacks a clear strategy for determining the optimal truncation filter threshold (γ\gammaγ) and when to activate the filter during training.
2. Experiments are conducted on a simplified regression problem, which may not fully represent real-world complexities, such as dynamic and high-dimensional datasets in healthcare or other sensitive applications.
3. Limited discussion of potential communication overhead in real-world FL settings, where client-server exchanges can be a bottleneck.
4. The method is focused on a specific form of UQ, and its generalizability to other machine learning tasks beyond regression remains uncertain.

Recommendation:
The paper presents a solid and novel contribution to federated learning with uncertainty quantification. However, due to the limitations in the experimental design and some unresolved methodological aspects, it is recommended for major revision before publication. Addressing the identified weaknesses and conducting more complex experiments would strengthen the study's impact.

---

### Official Review · Reviewer_E86V · 2024-09-26
**Two Students (2S) approach to enable uncertainty quantification (UQ) in federated learning (FL) clients.**

**Rating:** 7
**Confidence:** 3

**Review:**

This paper introduces two models, one focused on accurate predictions and another on quantifying uncertainty, using knowledge distillation from a Bayesian model ensemble (BME). It also incorporates a truncation filter to handle non-i.i.d. data by excluding unreliable client models based on credible intervals.
The paper is well-organized and clearly articulates the motivation behind the 2S approach. The methodology is described in detail. The idea of distilling a BME into two student models for the purpose of UQ in FL is novel and by introducing the truncation filter, the paper also contributes a new strategy for handling non-i.i.d. data in FL.

Pros:
- The truncation filter based on credible intervals is an effective strategy to mitigate the impact of non-i.i.d. data
- The manuscript is well-structured

Cons:
- The experiments are limited to a simple regression task
- The paper could benefit from a broader comparison to other recent methods in FL


Overall, the paper is a valuable addition to the workshop and opens avenues for future research.

---

### Decision · Program_Chairs · 2024-10-09

Accept (Poster)